# Involvement of Personality and Health Status in the Psychological Wellbeing of Subjects with Chronic Disease

**DOI:** 10.3390/bs14020099

**Published:** 2024-01-28

**Authors:** Cristina Rivera-Picón, Juan Luis Sánchez-González, Marta Rivera-Picón, Pedro Manuel Rodríguez-Muñoz

**Affiliations:** 1Faculty of Health Sciences, University of Castilla-La Mancha, 45660 Talavera de la Reina, Spain; 2Faculty of Nursing and Physiotherapy, University of Salamanca, 37007 Salamanca, Spain; 3University Hospital of Salamanca, 37007 Salamanca, Spain; mriverap@saludcastillayleon.es; 4Faculty of Health Sciences, University of Castilla-La Mancha, 45004 Toledo, Spain; pedrom.rodriguez@uclm.es; 5Department of Nursing, Instituto Maimónides de Investigación Biomédica, 14002 Córboda, Spain

**Keywords:** psychological wellbeing, personality, HIV, diabetes, chronic disease

## Abstract

(1) Background: Psychological wellbeing correlates with improved physical and psychological health, as this construct plays a fundamental role in disease recovery and health maintenance. Hence, for healthcare professionals, understanding the factors that predict psychological wellbeing is of great interest. Thus, the objective of this study was to determine whether health status and personality traits influence psychological wellbeing. (2) Methods: The total sample (N = 600) consisted of HIV patients, individuals with diabetes, and healthy subjects from the Salamanca Clinical Hospital. The instruments used for data collection included a sociodemographic questionnaire, Ryff’s Psychological Wellbeing Scale, and the Spanish version of the Big Five Taxonomy to measure personality. (3) Results: Specific personality traits, such as Emotional Stability, Extraversion, Responsibility, and Integrity were significant predictors of different dimensions of psychological wellbeing. Regarding health status, individuals with diabetes and healthy subjects, compared to HIV+ subjects, were associated with higher levels of psychological wellbeing dimensions. (4) Conclusions: Individual differences in personality traits and the diagnosis of a chronic condition may play a fundamental role in psychological wellbeing. These conclusions are of great interest for developing strategies aimed at individuals with chronic illnesses and specific personality traits associated with poorer psychological wellbeing.

## 1. Introduction

Psychological wellbeing has become a global concern, particularly in the context of chronic pathologies [1,2,3]. Chronic illnesses negatively impact the physical, mental, and social aspects of an individual’s life, posing a clear stress factor that threatens their wellbeing [4,5].

However, each illness entails specific changes that have a different impact on each individual’s wellbeing based on the pathology [6,7]. In this context, we emphasize HIV infection, a highly complex pathology in which individuals must cope with numerous physiological, sociocultural, economic, and psychological stressors. HIV-positive patients face social stigma, experience increased discrimination, and have reduced social support [8,9,10]. All of these factors can significantly affect the mental health of affected individuals.

Moreover, even in similar situations, the emotional reactions and adaptations of individuals can vary. Hence, several questions arose that motivated our study. Beyond the illness process, do other individual variables, such as personality, modulate the processes of psychological wellbeing in individuals?

Based on personality traits and psychological wellbeing, it is emphasized that conditions such as adopting an optimistic and positive outlook in the face of complications, attempting to derive positive aspects from problems and feeling fortunate, focusing on problem solving, planning, seeking social support, self-control, and self-esteem contribute to greater personal wellbeing. Conversely, feeling guilty or responsible for problems and difficulties, as well as avoiding problems, foster personal distress [11].

Over time, various approaches have been developed to study personality traits, primarily aiming to organize, group, and categorize them to determine variability and individual differences in personality. The most widely accepted and popular model in current psychology, both empirically and theoretically, is the “Big Five Model.” Hence, it is among the most widely employed and recognized personality models [12].

Regarding psychological wellbeing, Carol Ryff is one of the principal researchers in this field. Using her multidimensional approach, she delineated six dimensions to define and assess psychological wellbeing: self-acceptance, positive relationships, autonomy, environmental mastery, purpose in life, and personal growth.

Furthermore, we justify this study’s importance by noting that psychological wellbeing has been linked to better physical health and increased longevity. It is considered an indicator of positive functioning in individuals, enabling their personal development and growth [13,14]. Data have been found regarding the relationship between psychological wellbeing and biological indicators associated with health. The reason for this relationship is not clear, although it is suspected that psychological wellbeing is associated with medium- and long-term affective regulation mechanisms through the pursuit of survival behaviors and adaptation to environmental demands. Additionally, there is a relationship between positive affect and a lower risk of disease. Therefore, there is evidence that individuals experiencing more positive emotions are less likely to develop diseases such as upper respiratory tract infections, depression, arthritis, diabetes, and cancer.

Therefore, understanding the personality characteristics associated with poorer psychological wellbeing is crucial for focusing on effective and individualized health interventions. Thus, this study aimed to determine whether health status and personality traits influence psychological wellbeing.

## 2. Materials and Methods

### 2.1. Aim and Design of the Study

This study aimed to determine whether health status and personality traits influence psychological wellbeing. We conducted a non-experimental, cross-sectional study with a correlational objective.

### 2.2. Participants

As shown in Table 1, the final sample comprised 600 participants. Based on the participants’ health status, three categories were identified: HIV patients (N = 199), individuals with diabetes (N = 201), and healthy subjects (N = 200).

Participants were required to meet the following inclusion criteria. Individuals with HIV or diabetes had to have a confirmed diagnosis of the respective disease, regardless of its stage. For healthy subjects, the criteria did not include a diagnosis of chronic illness. All participants had to be of legal age and voluntarily participating in the study. Exclusion criteria included suffering from any medical or psychological condition that would hinder the participant from completing the study or from being able to sign an informed consent form.

Prior to data collection, the study obtained authorization from the University Assistance Complex of Salamanca and the Primary Care Management of Salamanca, and received a favorable report from the Ethics Committee “Research with Medicines in the Health Area of Salamanca,” with code CEIC: PI02/01/2018. Furthermore, participants’ participation was voluntary, and we ensured confidentiality and anonymity through the provision of an information sheet and requesting the completion of informed consent.

### 2.3. Data Collection

The sample was collected at the University Assistance Complex of Salamanca, specifically, at the University Hospital of Salamanca (HUS).

A subset of patients with HIV infection was obtained through incidental sampling. After obtaining the sample of HIV-positive individuals, samples of diabetic patients and healthy subjects were selected using quota sampling with equivalent age ranges, sexes, and educational levels, aiming to achieve homogeneous sub-samples. Thus, a quota sampling technique was employed for the two remaining subgroups to achieve homogeneous groups, thereby avoiding biases stemming from sociodemographic variables.

The instruments used for data collection were as follows.

#### 2.3.1. Sociodemographic Data Questionnaire

All participants were asked to provide sociodemographic information, including age, gender, marital status, and educational level. The participants’ health status was also recorded, specifying whether they had any pathology.

#### 2.3.2. Ryff’s Psychological Wellbeing Scale

The “Psychological Wellbeing Scale” (Scales of Psychological Wellbeing—SPWB) was developed by Ryff in 1989 and adapted by Van Dierendonck in 2004. Spanish adaptation of this scale was carried out by Díaz et al. (2006) [15].

The scale was developed to measure the dimensions of psychological wellbeing following Ryff’s model.

Table 2 displays the psychometric properties of the Spanish version of this scale, with 39 items (the original) and 29 items (the version proposed by Díaz et al.). Both versions included items written in both a direct and reverse manner. A total score can be obtained for each dimension.

In this research, the 29-item version was chosen for two reasons: (a) its brevity, (b) its psychometric properties, which yielded better results in the abbreviated version.

#### 2.3.3. Personality: Spanish Big Five Taxonomy

Th work of Iraegui and Quevedo-Aguado (2002) was used to assess personality factors. This research involved a psycholinguistic approach to the study of personality following the ‘Big Five hypothesis.’ A principal component factor analysis was applied to the 150 mini-factors identified in this research as personality descriptors. The Kaiser rule was used to select the number of factors to retain, and varimax normalization was employed as the rotation method [16].

The five-factor solution required ten iterations for convergence and explained 19.36% of the total variance, with a Cronbach’s α of 0.88. Table 3 displays the psychometric properties of each factor, as well as its correspondence with ‘McCrae and Costa’s Big Five model’. Factor 3 showed the greatest discrepancies in the original model.

For our study, a reduced scale of 50 personality descriptors has been created, with ten descriptors for each factor (five positive and five negative). These descriptors were chosen based on their correlations with the corresponding factor.

### 2.4. Data Analysis

Statistical analyses were conducted using Statistical Package for the Social Sciences (SPSS) version 25 from International Business Machines (IBM Corp., Armonk, NY, USA). Linear regression analysis was employed to address the proposed objective. This technique complies with five assumptions: independence, non-collinearity, linearity, homoscedasticity, and normality.

The assumption of independence was examined through the Durbin–Watson test, where values between 1.5 and 2.5 are considered appropriate. To assess the absence of collinearity, tolerance statistics and the variance inflation factor were employed. Lack of collinearity is considered when values are above 0.10 for the former and below 10 for the latter. The next two assumptions, linearity and homoscedasticity, were evaluated through graphical analysis of partial regression plots and scatter plots, respectively. Finally, the verification of the normality assumption was conducted both graphically and statistically, using the Kolmogorov–Smirnov test. Thus, after confirming the assumptions, linear regression analysis was conducted.

## 3. Results

### 3.1. Psychological Wellbeing: Self-Acceptance

The ANOVA resulted in a significant value for the test, rejecting H0 (F(gl) = 162.821(596), *p* < 0.001).

Regression analysis revealed that the best model included four variables, ($R^{2}$ = 0.579, $R_{A}^{2}$ = 0.576, F(gl1, gl2) = 8.661(2, 591), *p* = 0.025). The predictor variables included in the final model accounted for 57.9% of the variance in the DVs.

In Table 4, it can be observed that emotional stability, responsibility, and extraversion are positively related to the self-acceptance dimension of wellbeing. Personality Factor 1 (emotional stability) had the most significant weight (B = 0.186, β = 0.471, t = 11.501, *p* < 0.001). Regarding health status, both healthy and diabetic individuals had positive and significant coefficients.

### 3.2. Psychological Wellbeing: Positive Relationships

ANOVA resulted in a significant value for the test, rejecting H0 (F(df) = 117.851 (595), *p* < 0.001).

Regression analysis revealed that the best model included three variables ($R^{2}$ = 0.444, $R_{A}^{2}$ = 0.440, F(gl1, gl2) = 0.031 (1, 591), *p* < 0.001). The predictor variables included in the final model accounted for 44.4% of the variance in the DVs.

In Table 5, it can be observed that emotional stability and extraversion positively relate to the positive relationships dimension of psychological wellbeing. Personality Factor 1 (emotional stability) had the most significant weight (B = 0.197, β = 0.352, t = 8.764, *p* < 0.001). Both healthy subjects and diabetic individuals show a significant coefficient.

### 3.3. Psychological Wellbeing: Autonomy

ANOVA resulted in a significant value for the test, rejecting H0 (F(df) = 69.474 (596), *p* < 0.001).

Regression analysis revealed that the best model included four variables ($R^{2}=0$.319, $R_{A}^{2}$ = 0.315, F(gl1, gl2) = 4.373(1, 592), *p* = 0.037). These predictor variables included in the final model accounted for 31.9% of the variance in the DVs.

Table 6 shows that emotional stability, extraversion, and integrity are positively related to autonomy, while the agreeableness is negatively related to autonomy. Personality Factor 1 (emotional stability) had the most significant weight (B = 0.227, β = 0.389, t = 7.769, *p* < 0.001).

### 3.4. Psychological Wellbeing: Environmental Mastery

ANOVA resulted in a significant value for the test, rejecting H0 (F(df) = 152.435(596), *p* < 0.001).

Regression analysis revealed that the best model included three variables ($R^{2}=0$.507, $R_{A}^{2}$ = 0.504, F(gl1, gl2) = 9.007(2, 592), *p* < 0.001). The predictor variables included in the final model accounted for 50.7% of the variance in the DVs.

Table 7 shows that emotional stability and responsibility are positively related to the psychological wellbeing dimension of environmental mastery. Regarding health status, both healthy subjects and diabetic individuals show a significant coefficient in both cases.

### 3.5. Psychological Wellbeing: Personal Growth

ANOVA resulted in a significant result, rejecting H0 (F(df) = 88.891(596), *p* < 0.001).

Regression analysis revealed that the optimal model included four variables, ($R^{2}$ = 0.429, $R_{A}^{2}$ = 0.425, F(gl1, gl2) = 5.693(1, 591), *p* = 0.017). The predictor variables included in the final model accounted for 42.9% of the variance in DVs.

Table 8 shows that emotional stability, integrity, and extraversion are positively related to personal growth. Personality Factor 1 (emotional stability) had the most significant weight (B = 0.144, β = 0.406, t = 8.942, *p* < 0.001). Regarding health status, only the dummy variable for healthy subjects was significant (B = 0.970, β = 0.141, t = 3.795, *p* < 0.001).

### 3.6. Psychological Wellbeing: Purpose in Life

ANOVA resulted in a significant value for the test, rejecting H0 (F(df) = 212.027(596), *p* < 0.001).

Regression analysis revealed that the best model included three variables ($R^{2}$ = 0.589, $R_{A}^{2}$ = 0.586, F(df1, df2) = 0.039(1, 592), *p* < 0.001). The predictor variables included in the final model accounted for 58.9% of the variance in VDs.

Table 9 shows that emotional stability and responsibility are positively related to purpose in life. Regarding health status, both healthy and diabetic individuals had a significant coefficient.

## 4. Discussion

The maintenance and development of psychological wellbeing may be conditioned by various factors. Among them, we highlight individual differences in personality traits and the diagnosis of a chronic pathology.

Regarding Personality, as previously explained, the Spanish Five-Factor Model hierarchically organizes five personality factors: emotional stability, agreeableness, integrity, responsibility, and extraversion. Before discussing the obtained results, we will briefly outline the evidence with the intention of synthesizing the most important conclusions.

Concerning the psychological wellbeing variable, the personality factor emotional stability was positively related to all dimensions. Additionally, responsibility was associated with higher self-acceptance, environmental mastery, and purpose in life. Extraversion was positively related to self-acceptance, positive relationships, autonomy, and personal growth. Integrity was positively associated with autonomy and personal growth. The trait of agreeableness predicted lower levels of autonomy.

Moreover, the subjects’ health status was associated with all dimensions of psychological wellbeing, except for autonomy. Diabetic and healthy subjects, compared to HIV+ subjects, were related to higher levels in the dimensions of self-acceptance, positive relationships, environmental mastery, and purpose in life. Regarding personal growth, healthy subjects scored higher than HIV-positive individuals.

Thus, both health status and personality traits allowed us to predict the dimensions of psychological wellbeing most frequently used by the subjects.

If we focus on research analyzing personality traits that might predispose individuals to higher levels of psychological wellbeing, initial studies have argued that neuroticism and extraversion are the traits most closely related to psychological wellbeing [17]. Current research supports this relationship [18]. To compare this statement with our results, it is worth noting that individuals with low scores in neuroticism are positioned on the opposite side of the emotional stability factor. However, there are also more publications that, while acknowledging the strong relationship of these factors with wellbeing, add other traits related to this construct [19]. For instance, Meléndez et al. (2019) [20] argued that, besides neuroticism and extraversion, responsibility was also a trait highly related to psychological wellbeing. Neuroticism was negatively associated with all dimensions of psychological wellbeing, whereas extraversion and responsibility were positively associated. Delhom et al.’s study (2019) obtained results similar to ours, adding Integrity as an important predictor, showing a positive association with personal growth [21]. The study by Barra et al. (2013) also confirmed these results, emphasizing the strong influence of personality traits on wellbeing levels [22]. Furthermore, these projects suggested that neuroticism, or its opposite factor, emotional stability, is one of the factors most related to psychological wellbeing, which is consistent with the results of our work and those of other studies [23,24,25,26,27,28,29].

Thus, the evidence from studies that have examined psychological wellbeing in subjects with certain chronic diseases resembles the results of our study. Another example is the study by García-Viniegras and González (2007), who analyzed the wellbeing of breast cancer patients. This project highlighted that self-confidence, emotional stability, agreeableness, and self-esteem were associated with higher levels of wellbeing [30].

However, the research that has examined psychological wellbeing in individuals with certain chronic illnesses has primarily focused on the elements that can influence or modulate such wellbeing. On the other hand, most studies on psychological wellbeing in chronic patients concentrate on demonstrating the impact of psychological wellbeing on these individuals. An example of this is the study by Vázquez et al. on inflammatory processes, which determined that individuals with higher levels of interpersonal wellbeing and a sense of purpose in life exhibit lower levels of IL-6 [31]. Moreover, Carol Ryff, based on some of her research, has found relevant results in this area. In samples of older women, she has determined that those with better interpersonal relationships, a greater sense of personal growth, and higher levels of purpose in life present a lower cardiovascular risk, indicated by lower levels of glycosylated hemoglobin, lower body weight, higher rates of HDL cholesterol, and better endocrine regulation [32].

Therefore, regarding health status, fewer studies have focused on predicting levels of psychological wellbeing based on subjects’ health status. However, our results are consistent with those published by other authors, such as Chongwo et al. (2018) and Ramkisson et al. (2016), who showed that the absence of illness and the diagnosis of diabetes are associated with higher levels of psychological wellbeing than those found in HIV subjects [33,34].

Due to the limited publications on psychological wellbeing in chronic illness and its implications for health, we consider the study of this topic to be of great importance.

Limitations: The literature is limited on the impact of diabetes and HIV diagnosis on the psychological wellbeing of subjects. This prevents the discussion and comparison of our results. Another limitation is the exclusion of sociodemographic variables such as gender, which could help predict patients’ psychological wellbeing. Another limitation of the present study is the sampling method: non-random sampling. This diminishes the external validity of the study, making it challenging to generalize the results. However, given the characteristics of the sample to which we had access, conducting random sampling was impractical. Finally, we emphasize that the severity and prognosis of the disease may have also influenced the outcomes. Future research should take into account the limitations mentioned earlier to provide more specific evidence regarding the results.

## 5. Conclusions

In conclusion, both individual differences in personality traits and the diagnosis of a chronic condition can significantly impact the development and maintenance of psychological wellbeing. specifically, emotional stability, extraversion, responsibility, and integrity were significant predictors of the different dimensions of psychological wellbeing. The health status of the subjects was associated with all dimensions of psychological wellbeing, except for autonomy. Diabetic and healthy subjects, in comparison to HIV-positive individuals, were related to higher levels in the dimensions of self-acceptance, positive relationships with others, environmental mastery, and purpose in life. In the dimension of personal growth, healthy subjects scored higher than HIV-positive subjects. Thus, we can confirm that diabetic and healthy subjects exhibited higher levels of psychological wellbeing compared to individuals with HIV. These conclusions may be justified by the fact that HIV is a stigmatizing and contagious condition compared with diabetes, thereby correlating with poorer levels of the studied psychological variables.

These findings hold significant importance in clinical practice, as they could assist in forecasting and anticipating lower levels of psychological wellbeing in individuals at higher risk based on their personality characteristics and the chronic conditions that they endure.

Considering the findings in conjunction with the HIV-associated stigma, future research endeavors could benefit from an exploration of the transmission methods and sexual identities of individuals living with HIV. Additionally, it would also be important to consider the limitations presented in this research. Therefore, we consider it important to conduct further studies that include sociodemographic variables, as they play a significant role in determining psychological wellbeing. Finally, considering these considerations, we aim to expand the research to encompass more chronic conditions, with the goal of facilitating tailored interventions for each specific ailment.

## Figures and Tables

**Table 1 behavsci-14-00099-t001:** Sociodemographic variables based on health status.

	HIV		Diabetes		Healthy Subjects		Total	$\chi^{2}$	ET	*p*
	N	%	N	%	N	%	N			
N° participants	199	33.2%	201	33.5%	200	33.3%	600			
Sex										
Woman	48	24.1%	58	28.9%	58	29.0%	164	1.548	0.051	0.461
Man	151	75.9%	143	71.1%	142	71.0%	436			
Marital status										
Married/couple	58	29.1%	123	61.2%	124	62.0%	305	63.417	0.230	<0.001
Single/widowed/other	117	58.8%	64	31.8%	51	25.5%	232			
Separated/divorced	24	12.1%	14	7.0%	25	12.5%	63			
Level of studies										
Secondary or lower	168	84.4%	160	79.6%	156	78.0%	484	2.858	0.069	0.240
Superior	31	15.6%	41	20.4%	44	22.0%	116			
Age										
43 years or younger	50	25.1%	46	22.9%	59	29.5%	155	10.930	0.095	0.091
44 to 50 years	61	30.7%	63	31.3%	55	27.5%	179			
51 to 55 years	57	28.6%	44	21.9%	38	19.0%	139			
56 years or older	31	15.6%	48	23.9%	48	24.0%	127			

N: number of subjects, %: percentage, $\chi^{2}$: chi-squared, ET: effect size, *p*: *p*-value.

**Table 2 behavsci-14-00099-t002:** Psychometric properties of the Psychological Wellbeing Scale (Spanish version).

	39-Item Version		29-Item Version	
	N° Items	α Cronbach	N° Items	α Cronbach
Self-acceptance	6	0.83	4	0.84
Positive relations	6	0.81	6	0.78
Autonomy	8	0.73	6	0.70
Environmental mastery	6	0.71	5	0.82
Purpose in life	7	0.83	5	0.70
Personal growth	6	0.68	4	0.71

**Table 3 behavsci-14-00099-t003:** Psychometric properties of personality dimensions.

Found Factor	Explained Variance	Correspondence
F1. Emotional Stability	4.21%	F4. Neuroticism
F2. Agreeableness	4.15%	F2. Disagreeableness
F3. Integrity	4.00%	F5. Openness
F4. Responsibility	3.64%	F3. Conscientiousness
F5. Extraversion	3.20%	F1. Extroversion

**Table 4 behavsci-14-00099-t004:** Final model coefficients: PWB—self-acceptance.

	B	E.T.	* **β** *	t	*p*
Constant	1.270	0.443		2.866	0.004
F1 Emotional Stability Factor	0.186	0.016	0.471	11.501	<0.001
F4 Responsibility Factor	0.103	0.018	0.207	5.618	<0.001
F5 Extraversion Factor	0.067	0.017	0.138	3.919	<0.001
Diabetes	0.760	0.247	0.099	3.077	0.002
Healthy subjects	1.004	0.250	0.131	4.022	<0.001

B, unstandardized regression coefficients; E.T., standard error; β, standardized regression coefficients; *p*, *p*-value.

**Table 5 behavsci-14-00099-t005:** Final model coefficients: PWB—positive relationships.

	B	E.T.	*β*	t	*p*
Constant	2.136	0.595		3.591	<0.001
F1 Emotional Stability	0.197	0.022	0.352	8.764	<0.001
F5 Extraversion	0.160	0.028	0.233	5.737	<0.001
Diabetes	3.229	0.397	0.298	8.136	<0.001
Healthy subjects	3.084	0.393	0.284	7.853	<0.001

B, unstandardized regression coefficients; E.T., standard error; β, standardized regression coefficients; *p*, *p*-value.

**Table 6 behavsci-14-00099-t006:** Final model coefficients: PWB—autonomy.

	B	E.T.	β	t	*p*
Constant	9.084	1.423		6.384	<0.001
F1 Emotional Stability	0.227	0.029	0.389	7.769	<0.001
F5 Extraversion	0.125	0.032	0.175	3.929	<0.001
F3 Integrity	0.127	0.042	0.140	2.985	0.003
F2 Agreeableness Factor	−0.076	0.036	−0.092	−2.091	0.037

B, unstandardized regression coefficients; E.T., standard error; β, standardized regression coefficients; *p*, *p*-value.

**Table 7 behavsci-14-00099-t007:** Final model coefficients: PWB—environmental mastery.

	B	E.T.	β	t	*p*
Constant	3.606	0.491		7.348	<0.001
F1 Emotional Stability	0.222	0.017	0.503	13.071	<0.001
F4 Responsibility	0.122	0.022	0.220	5.530	<0.001
Diabetes	1.247	0.296	0.146	4.029	<0.001
Healthy Subjects	0.812	0.299	0.095	2.717	0.007

B, unstandardized regression coefficients; E.T., standard error; β, standardized regression coefficients; *p*, *p*-value.

**Table 8 behavsci-14-00099-t008:** Final model coefficients: PWB—personal growth.

	B	E.T.	β	t	*p*
Constant	7.871	0.591		13.307	<0.001
F1 Emotional Stability	0.144	0.016	0.406	8.941	<0.001
F3 Integrity	0.115	0.022	0.209	5.278	<0.001
F5 Extraversion	0.043	0.018	0.098	2.386	0.017
Diabetes	0.310	0.255	0.045	1.217	0.224
Healthy Subjects	0.970	0.256	0.141	3.795	<0.001

B, unstandardized regression coefficients; E.T., standard error; β, standardized regression coefficients; *p*, *p*-value.

**Table 9 behavsci-14-00099-t009:** Final model coefficients: PWB—purpose in life.

	B	E.T.	β	t	*p*
Constant	1516	0.516		2.941	0.003
F1 Emotional Stability	0.241	0.018	0.475	13.496	<0.001
F4 Responsibility	0.174	0.023	0.272	7.512	<0.001
Diabetes	1.969	0.311	0.200	6.322	<0.001
Healthy Subjects	2.152	0.314	0.219	6.849	<0.001

B, unstandardized regression coefficients; E.T., standard error; β, standardized regression coefficients; *p*, *p*-value.

## Data Availability

The data that support the findings of this study are available on request from the corresponding author. The data are not publicly available due to privacy or ethical restriction.

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
