# Peer review of "Involvement of Personality and Health Status in the Psychological Wellbeing of Subjects with Chronic Disease"

_behavsci, 2024, doi:10.3390/bs14020099_

Round 1

Reviewer 1 Report

Comments and Suggestions for Authors

The objective of this manuscript is to show individual difference in personality traits and type of chronic health condition play a significant role in the psychological well-being of the patient. This study compares HIV and diabetes as two chronic diseases along with healthy controls.

1. Overall, the manuscript is detailed but does not adequately prove the authors premise because of the complexity of HIV patients. The data is valuable, but the conclusions should be more specific to the primary population studied.

2. The authors obtained a subset of patients with HIV and used quota sampling to match healthy controls and patients with diabetes. this method of data collection does not negate the fact that the overall population was predominantly male. The rationale for using this method should be explained and this is a limitation that should be addressed. 

3. Although the authors list the limitation of not obtaining socioeconomic factors, these are very important in determining psychological well-being. This is a flaw in the methodology. 

4. The authors state that HIV was chosen because of its stigma of the disease. It would have been good to including the method of transmission and sexual identity of the participants. This would have allowed the researchers to measure more of HIV as a chronic disease and less of the additional factors that accompany this disease (which were clearly mentioned in the introduction).  

5. Disease severity and prognosis may have also affected the results.

Author Response

The authors would like to thank you for giving us the opportunity to revise and improve our manuscript; we also thank the reviewers for their thoughtful and constructive comments. We have considered the suggestions and have incorporated them into the revised manuscript. We believe that our manuscript is stronger as result of these modifications. An itemized point-by-point response to the reviewers’ comments is presented below written in red. The modifications in the manuscript are made in red:

Response to Reviewer 1 Comments

The objective of this manuscript is to show individual difference in personality traits and type of chronic health condition play a significant role in the psychological well-being of the patient. This study compares HIV and diabetes as two chronic diseases along with healthy controls.

  1. Overall, the manuscript is detailed but does not adequately prove the authors premise because of the complexity of HIV patients. The data is valuable, but the conclusions should be more specific to the primary population studied.

Thank you very much for the considerations. The information has been completed in the conclusions.

Line 322- line 330: “The health status of the subjects was associated with all dimensions of Psychological Well-being, except for Autonomy. Diabetic and healthy subjects, in comparison to HIV-positive individuals, were related to higher levels in the dimensions of Self-acceptance, Positive relations with others, Environmental mastery, and Purpose in life. In the dimension of Personal growth, healthy subjects scored higher than HIV-positive subjects. Thus, we can confirm that diabetic and healthy subjects exhibited higher levels of psychological well-being compared to individuals with HIV. These conclusions may be justified by the fact that HIV is a much more stigmatizing and contagious condition than diabetes, thereby correlating with poorer levels of the studied psychological variables”.

  1. The authors obtained a subset of patients with HIV and used quota sampling to match healthy controls and patients with diabetes. this method of data collection does not negate the fact that the overall population was predominantly male. The rationale for using this method should be explained and this is a limitation that should be addressed. 

Thank you so much for the contributions received.

Our intention was to obtain 3 homogeneous subsamples in terms of sociodemographic variables. In this way, we were not interested in finding differences between groups because, in the case that differences emerge, it would have been difficult to determine whether our results are due to the variable of interest or to the different samples. For example, if the HIV and Diabetes samples were different in terms of gender, we might raise the question: Do the differences found in Psychological Well-being really stem from the health condition, or could they be attributed to gender (e.g., higher psychological well-being in diabetics due to a larger number of women). "Therefore, we used quota sampling to obtain homogeneous groups that were later compared, thus avoiding biases stemming from sociodemographic variables. We specified this information in the lines 107-110. However, we acknowledge the limitation of employing a non-random sampling method, and we have added it as a limitation:

Line 310-314: “Another limitation of the present study is the sampling method: non-random sampling. This diminishes the external validity of the study, making it challenging to generalize the results. However, given the characteristics of the sample to which we had access, conducting random sampling was impractical.”

  1. Although the authors list the limitation of not obtaining socioeconomic factors, these are very important in determining psychological well-being. This is a flaw in the methodology. 

Thank you very much for the contributions. We justify this limitation by stating that the main focus of the article was to assess the relationship between psychological well-being and personality variables in patients with chronic illnesses. Currently, there is ample research on sociodemographic variables and their relation to these psychological variables, so our objectives and primary interest were different. Therefore, we aimed to obtain three homogeneous subsamples concerning sociodemographic variables, thus avoiding biases stemming from these factors. However, we acknowledge your consideration, and it would be crucial to include these socioeconomic factors in future articles. Therefore, we have included it as a future line of research:

Lines 337-340: “Additionally, it would also be important to consider the limitations presented in this research. Therefore, we consider it important to conduct further studies that include sociodemographic variables, as they play a significant role in determining psychological well-being”.

  1. The authors state that HIV was chosen because of its stigma of the disease. It would have been good to including the method of transmission and sexual identity of the participants. This would have allowed the researchers to measure more of HIV as a chronic disease and less of the additional factors that accompany this disease (which were clearly mentioned in the introduction).  

We appreciate your suggestion and have included it in the article as one of our proposed future directions. We agree and believe that exploring the transmission and sexual identity of the participants could be very interesting

Lines 335-337: Considering the findings in conjunction with the HIV-associated stigma, future research endeavors could benefit from an exploration of the transmission methods and sexual identities of individuals living with HIV .

  1. Disease severity and prognosis may have also affected the results.

We appreciate your consideration. It is included as one of the study limitations:

Lines 314-316: Finally, we emphasize that the severity and prognosis of the disease may have also influenced the outcomes. Future research should take into account the limitations mentioned earlier to provide more specific evidence regarding the found results.

Reviewer 2 Report

Comments and Suggestions for Authors

Thank you for this interesting manuscript. I have a few comments:

* the text throughout mentions the term "Responsibility", but this word does not appear in the  relevant tables. IN stead, in Tables 4, 7 & 8 it appears to be represented by F4 which is termed "conscientiousness". If you agree, this needs correcting.

* I note that the  group of HIV patients appears different from the Diabetes and healthy patients  eg (from the Discussion) Diabetic and healthy subjects, compared to 223 HIV+ subjects, were related to higher levels in the dimensions of Self-acceptance, Positive 224 Relationships, Environmental Mastery, and Purpose in Life. Regarding Personal Growth, 225 healthy subjects scored higher than HIV individuals. I further note that the sampling for the HIV + patients was incidental, as opposed to the other two groups, who were collected through quota sampling. Could this different selection method, which would make the HIV+ sample non-random, have affected the results? I believe this should be included in the Discussion or limitations section of the paper. 

Comments on the Quality of English Language

Some minor changes needed eg in introduction para 3 line 42/3 the following sentence appears not to warrant a question mark:  Beyond the  illness process, other individual variables such as personality modulate the processes of  psychological well-being in individuals?

Alternatively, inserting the word "do" after the comma in that sentence woudl correct the english.

Author Response

The authors would like to thank you for giving us the opportunity to revise and improve our manuscript; we also thank the reviewers for their thoughtful and constructive comments. We have considered the suggestions and have incorporated them into the revised manuscript. We believe that our manuscript is stronger as result of these modifications. An itemized point-by-point response to the reviewers’ comments is presented below written in red. The modifications in the manuscript are made in red:

Response to Reviewer 2 Comments

Gracias por este interesante manuscrito. Tengo algunos comentarios:

* En todo el texto se menciona el término "Responsabilidad", pero esta palabra no aparece en los cuadros correspondientes. En cambio, en las Tablas 4, 7 y 8 parece estar representado por F4, que se denomina "escrupulosidad". Si está de acuerdo, esto debe corregirse.

Thank you very much for your observation. It is an error. We have corrected the term in all the erroneous tables. Thank you very much.

* Observo que el grupo de pacientes con VIH parece diferente del de Diabetes y los pacientes sanos por ejemplo (de la Discusión) Los sujetos diabéticos y sanos, en comparación con 223 sujetos VIH+, se relacionaron con niveles más altos en las dimensiones de Autoaceptación, Relaciones Positivas 224, Dominio del Entorno y Propósito en la Vida. En cuanto al Crecimiento Personal, 225 sujetos sanos puntuaron más alto que los individuos con VIH. Observo además que el muestreo para los pacientes VIH+ fue incidental, a diferencia de los otros dos grupos, que se recogieron a través de un muestreo por cuotas. ¿Podría este método de selección diferente, que haría que la muestra VIH+ no fuera aleatoria, haber afectado los resultados? Creo que esto debería incluirse en la sección de Discusión o limitaciones del documento.

Thank you very much for your consideration; we find it highly relevant and have added a new limitation based on your reflection.

The reason for selecting HIV+ subjects through incidental sampling and subsequently selecting healthy subjects and diabetic subjects through quota sampling is justified below:

Our intention was to obtain three homogeneous subsamples in terms of sociodemographic variables. In this way, we were not interested in finding differences between groups because, in the case that differences emerged, it would have been difficult to determine whether our results are due to the variable of interest or to the different samples. For example, if the samples of HIV and Diabetes were different in terms of gender, we might raise the question: Are the differences found in Psychological Well-being really due to the health condition, or could they be attributed to gender (e.g., higher psychological well-being in diabetics due to a higher number of women)? Therefore, we used quota sampling to obtain homogeneous groups that were later compared, thus avoiding biases stemming from sociodemographic variables. We specified this information in the lines 107-110. However, we acknowledge the limitation of employing a non-random sampling method, and we have added it as a limitation:

Line 310-314: “Another limitation of the present study is the sampling method: non-random sampling. This diminishes the external validity of the study, making it challenging to generalize the results. However, given the characteristics of the sample to which we had access, conducting random sampling was impractical.”

Comentarios sobre la calidad de la lengua inglesa

Se necesitan algunos cambios menores, por ejemplo, en la introducción, párrafo 3, línea 42/3, la siguiente oración parece no justificar un signo de interrogación: ¿Más allá del proceso de enfermedad, otras variables individuales, como la personalidad, modulan los procesos de bienestar psicológico en los individuos?

Alternativamente, insertar la palabra "do" después de la coma en esa oración corregiría el inglés.

Thank you very much for your consideration. Indeed, inserting the word 'do' after the comma in that sentence would correct the English. We have taken your suggestion into account and included it in line 45.

Round 2

Reviewer 1 Report

Comments and Suggestions for Authors

The authors have adequately addressed the previous comments and concerns.